# HELLP Syndrome—Holistic Insight into Pathophysiology

**DOI:** 10.3390/medicina58020326

**Published:** 2022-02-21

**Authors:** Aida Petca, Bianca Corina Miron, Irina Pacu, Mihai Cristian Dumitrașcu, Claudia Mehedințu, Florica Șandru, Răzvan-Cosmin Petca, Ioana Cristina Rotar

**Affiliations:** 1“Carol Davila” University of Medicine and Pharmacy, 050474 Bucharest, Romania; aida.petca@umfcd.ro (A.P.); mihai.dumitrascu@umfcd.ro (M.C.D.); claudia.mehedintu@umfcd.ro (C.M.); florica.sandru@umfcd.ro (F.Ș.); razvan.petca@umfcd.ro (R.-C.P.); 2Department of Obstetrics and Gynecology, Elias University Hospital, 011461 Bucharest, Romania; 3Department of Obstetrics and Gynecology, “Sf. Pantelimon” Emergency Clinical Hospital, 021623 Bucharest, Romania; 4Department of Obstetrics and Gynecology, University Emergency Hospital, 050098 Bucharest, Romania; 5Department of Obstetrics and Gynecology, Malaxa Clinical Hospital, 022441 Bucharest, Romania; 6Department of Dermatology, Elias University Emergency Hospital, 011461 Bucharest, Romania; 7Department of Urology, “Prof. Dr. Th. Burghele” Clinical Hospital, 050659 Bucharest, Romania; 8“Iuliu Hatieganu” University of Medicine and Pharmacy, 400012 Cluj-Napoca, Romania; cristina.rotar@umfcluj.ro; 9Department of Obstetrics and Gynecology, Emergency Clinical County Hospital Cluj-Napoca, 400006 Cluj-Napoca, Romania

**Keywords:** HELLP, pathogenesis, microangiopathy, genetic, placenta, DIC

## Abstract

HELLP syndrome, also known as the syndrome of hemolysis, elevated liver enzymes, and low platelets, represents a severe pregnancy complication typically associated with hypertension. It is associated with increased risks of adverse complications for both mother and fetus. HELLP occurs in 0.2–0.8% of pregnancies, and, in 70–80% of cases, it coexists with preeclampsia (PE). Both of these conditions show a familial tendency. A woman with a history of HELLP pregnancy is at high risk for developing this entity in subsequent pregnancies. We cannot nominate a single worldwide genetic cause for the increased risk of HELLP. Combinations of multiple gene variants, each with a moderate risk, with concurrent maternal and environmental factors are thought to be the etiological mechanisms. This review highlights the significant role of understanding the underlying pathophysiological mechanism of HELLP syndrome. A better knowledge of the disease’s course supports early detection, an accurate diagnosis, and proper management of this life-threatening condition.

## 1. Introduction

Hemolysis, elevated liver enzymes, and low platelet count syndrome, otherwise known as HELLP syndrome, is considered a severe complication of pregnancy that coexists in 70–80% of cases with preeclampsia [1]. The risk of developing HELLP syndrome within a pregnancy ranges between 0.2% and 0.8%, with a 0–24% mortality rate [2,3,4]. The perinatal death rate goes up to 37% [2,3]. It is associated with an increased risk of severe complications for both mother and offspring [4,5,6,7].

Usually, patients develop HELLP syndrome before 36 weeks of gestation with vague symptoms such as malaise (90%), right-upper-quadrant pain (90%), nausea, or vomiting [2].

The precise etiology of HELLP syndrome is not fully understood. Current hypotheses include genetic mutation (both maternal and fetal) and inflammatory origin [3]. HELLP syndrome implies impaired placentation during the early stages of pregnancy, associated with hepatic and coagulation cascade involvement [3]. Recent studies focused on the implication of placenta-derived inflammatory cytokines and the role of immune maladaptation in HELLP pathogenesis [6].

A genetic analysis of the inheritability predisposition of HELLP syndrome in pregnancy has been explored. Results revealed both genetic and immunological factors that play a role in pathogenesis [3].

Studies have shown that sisters and children of a woman who develops HELLP syndrome have an increased risk for this condition [6,8]. A woman with a previous HELLP pregnancy is at high risk for developing HELLP and PE in subsequent pregnancies [9,10,11,12]. A first pregnancy is less probable to be associated with increased risk for HELLP but has a considerably higher risk of PE [4]. The antiphospholipid antibody syndrome (APLS) may be related to the early onset of HELLP [6].

The classification of HELLP syndrome is based on two main diagnostic definitions. The Tennessee classification is widely used for diagnosis [13]. It requires the presence of (1) microangiopathic hemolytic anemia with abnormal blood smear and low serum haptoglobin, (2) elevated LDH levels above 600 IU/L and ASAT above 70 IU/L (both enzyme levels more than twofold the upper limit of normal values) or bilirubin more than 1.2 mg/dL, and (3) a platelet count below 100 × 10^9^ L^−1^. The Mississippi classification underlines the severity of the disorder according to the nadir of the platelet count [14]. Current literature describes a clinically less severe entity, referred to as incomplete HELLP syndrome, with only two criteria (“ELLP”) [6].

The onset of HELLP syndrome before 28 weeks’ gestation accounts for about 20–30% of the cases and is associated with severe disease with rapid onset of clinical manifestations that often coexist with fetal growth restriction [6,15]. Fetus delivery is the only efficient treatment of HELLP syndrome [16].

Cytokines are involved in many pathological processes, including cancer and cardiovascular diseases [4]. So far, recent literature has demonstrated their role in innate immunity, cell growth, angiogenesis, apoptosis, and differentiation [4,5,6,7,8,17,18]. Although the precise reason has not yet been fully elucidated, inappropriate levels of cytokines that lead to a series of pathophysiological modifications seem to be the base of HELLP syndrome [4,5,6,7,8].

The main purpose of this review is to determine which placental-derived factors are potentially toxic for liver sinusoidal endothelial cells (LSECs). Last but not least, we aim to unravel the pathogenic mechanisms of endothelial damage and activation of inflammatory reactions in the liver.

## 2. Placental Pathogenesis of HELLP Syndrome

Considered to have a role in the pathogenesis of HELLP syndrome is an inflammatory response in the placenta–liver axis [4]. The hepatic function in HELLP patients improves within 6 weeks of delivery which advocates regarding the placenta as the primary situs involved in the malfunctioning of the liver [17]. The placental factors and vasoactive substances induce an enhanced inflammatory state and endothelial damage (Figure 1), resulting in sinusoidal obstruction syndrome (SOS) [18]. The endothelial damage in the liver is thought to be responsible for the obstruction with red blood cells of the Disse space, an area between liver sinusoidal endothelial cells and hepatocytes [4,19]. This leads to an increased formation of microthrombi, ischemia of the hepatocytes, and ultimately liver failure [4].

### 2.1. Placenta-Derived Factors Involved in the Pathophysiology of HELLP Syndrome

The factors involved in hepatic injury by increasing the apoptosis of the LSECs are elements implicated in vascular homeostasis, growth factors, and components involved in apoptosis (Figure 1 and Table 1).

These molecules are held accountable for promoting vasoconstriction by decreasing NO production, activating platelets with a shift in hemostasis, and activating LSECs favored by an increased inflammatory state [4]. Apoptosis of LSECs causes red blood cells to enter the space of Disse, where they cause obstruction and eventually lead to liver failure [4,18].

#### 2.1.1. Factors Influencing the Vasculature

VEGF (vascular endothelial growth factor)The vascular endothelial growth factor (VEGF) represents a major proangiogenic factor. Recent studies have shown that VEGF serum concentration, placental mRNA levels, and cytotrophoblast expression of VEGF and their receptors in HELLP patients are significantly lower than the serum concentration of patients suffering from severe preeclampsia (PE) but higher than the level in nonpregnant women [20,21]. Soluble vascular endothelial growth factor receptor-1 (sVEGFR-1), also known as soluble fms-like tyrosine kinase-1 (sFlt-1), secretion is increased in HELLP patients, preventing the interaction between VEGF and platelet growth factor with their receptors [22,23,24,25,26]. sFlt-1 is produced by the placenta, macrophages, endothelial cells, and vascular smooth muscle cells [21]. New studies cite the existence of a splice variant of sFlt (sFlt-e15a/sFlt-14), which is upregulated in the placenta of PE and HELLP patients [4,24].sENG (soluble endoglin)The concentration of soluble endoglin (sEng) is observed to be higher in women with HELLP syndrome compared with nonpregnant and preeclamptic women, as well as higher than the sFlt1 serum level [4,6]. It has been shown that both sEng and sFlt1 inhibit endothelial tube formation and, thus, damage endothelial integrity, resulting in vascular damage and leak [6,26]. Endoglin is a coreceptor for TGF-β1 and TGF-β3 isoforms, and it reduces TGF-β1 binding to receptor type II (TβRII) on endothelial cells. TGF- β1 is responsible for VEGF production by the stellate cells in the space of Disse. Therefore, sEng is thought to determine impaired production of VEGF by the pericytes [4,47,48,49].Gal-1 (galectin-1)Galectin-1 plays an important role in immune modulation and angiogenesis by binding to neuropilin-1 (NRP-1), which promotes activation and signaling of VEGFR2 [27]. Studies observed a reduced level of sFlt-1 after Gal-1 supplementation and, therefore, increased VEGF bioavailability. Gal-1 serum level is significantly increased in HELLP women [28,29]. Its expression is upregulated in the placenta’s syncytiotrophoblast. A negative correlation was noted between systemic Gal-1 levels and platelet counts in early-onset HELLP patients through a Ca^2+^-dependent mechanism [4]. Ca^2+^ levels are elevated on platelet stimulation with Gal-1, which signals thromboxane A2 (TXA2) synthesis [50].Gal-1 activates the P-selectin and GPIIIa expression dose-dependently, which triggers conformational changes in GPIIb/IIIa and F-actin polymerization on human platelets [29]. An upregulation in P-selectin expression on the activated platelets’ surface initiates a torrent of intracellular events in leukocytes and platelets. This promotes vascular inflammation and facilitates atherosclerosis and thrombotic episodes [4,29]. Incubating platelets with PGI2 or NO before exposure to Gal-1 prevents the shredding of microvesicles and P-selectin [29]. A complete blockage of Gal-1-induced P selectin and GPIIIa upregulation in disaccharide lactose treatment was reported [4].ET-1 (endothelin-1)Endothelin 1 binding to the endothelin A receptor (ET_A_) activates the ET pathway [30]. It mediates vasoconstriction, elevates blood pressure, contributes to oxidative stress, and increases inflammatory cytokines and CD4^+^ cells [30]. Circulating endothelin 1 level is elevated in HELLP patients. ET-1 dysfunction mediators are inflammatory cytokines, agonistic autoantibodies to the angiotensin II type 1 receptor, and increased sFlt1 and sEng [4].Angs-2 (angiopoietin-2)Angiopoietins play an essential role in promoting angiogenesis and maintaining vascular integrity [51]. Angiopoietin-1 (Angs-1) and angiopoietin-2 (Angs-2) bind to the same endothelial cell-specific tyrosine kinase receptor, which is activated by Angs-1 and blocked by Angs-2 [51]. Angs-1 is a promoter of endothelial cell survival, whereas Angs-2 is an activator of the endothelium [52]. Significantly higher Angs-2 levels were measured in HELLP patients in comparison to normal or preeclamptic pregnant women [53]. Angs-1 is considerably increased in HELLP patients compared to normal pregnant women [4].ADMA (asymmetric dimethylarginine)Asymmetric dimethylarginine represents an inhibitor of the enzyme NO synthase [33]. ADMA decreases the availability of NO, causing vasodilation [33]. Dimethylarginine dimethylaminohydrolase (DDHA) is an ADMA-degrading enzyme located in the placenta tissue [4]. Therefore, placental DDHA dysfunction is considered to be one of the major events implicated in PE and HELLP syndrome development [44,54]. ADMA serum concentrations are significantly higher in HELLP patients [4].

#### 2.1.2. Growth Factors

Activin and Inhibin

Inhibins and activins are dimeric disulfide-linked glycoproteins, members of the transforming growth factor-beta (TGF-β) family of cytokines [55]. Inhibins are heterodimers made of one α-subunit and one β-subunit, which can be either βA or βB subunits. There are three possible isoforms of activin: activin A (βA–βA), activin B (βB–βB), and activin AB (βA–βB) [56]. Recent studies have reported a significant synergic influence of activin A and VEGF on the survival of liver sinusoidal endothelial cells [37,38]. VEGF stimulates the production of activin A, thus amplifying the expression of VEGF receptors and increasing VEGF action. Inhibin A appears to be a low-affinity antagonist of activin A [57]. A stable complex that inhibits with high affinity is formed when inhibin A is combined with beta glycan [57].

Activin B is considered an important regulator of the gestation duration and onset of the parturition [40]. Activin B deficiency leads to failed initiation of the labor, which potentially results in the sickness of the pregnant woman and death of the fetus in utero [40]. Activin B enhances follicle-stimulating hormone (FSH) release, whereas inhibin B determines a decrease in FSH [4]. The fetoplacental unit, formed by the placental trophoblast, decidua, and fetal membranes, represents the primary source of circulating activin and inhibin [39]. There is an increase in inhibin A production in the extravillous trophoblast cells of HELLP and PE patients [35,58]. Studies have shown an elevated activin B serum level in the syncytiotrophoblast cells in HELLP syndrome rather than inhibin B [4,58].

The current literature states the variable effects of activin on trophoblast cells depending on gestation age. Spiral artery remodeling benefits from the proinvasive effects of these molecules exercised in pregnancy as early as 10 weeks gestation age. In later stages of pregnancy, proapoptotic effects are more frequent, along with PE [59].

#### 2.1.3. Apoptosis/Necrosis-Related Factors

Fas/FasL (Fas receptor/Fas ligand)

Fas receptor (Fas) and Fas ligand (FasL) are part of the TNF receptor family and regulate the inflammatory response via activation and proliferation of CD4^+^ T lymphocytes [60]. They can activate the apoptosis pathway [4]. Apoptosis is induced when FasL binds to Fas-positive cells through the activation of the extrinsic pathway or CD4^+^ cells [60]. Apoptosis, proliferation, and FasL expression in villous trophoblast of HELLP patients are notably higher than in normal pregnant women or PE [42,43]. Liver sinusoidal endothelial cells exhibit an increased expression of Fas [60]. The liver represents a physiological source of FasL [42,43]. Other sources are the placenta and the cytotoxic T lymphocytes [42,43]. Expression of FasL was not detected in the liver endothelial cells [43,61]. In HELLP syndrome, the FasL coming from the placenta binds Fas, thus causing apoptotic cell death in the liver of these women [42]. Exposing the liver sinusoidal endothelial cells to an increased TNFα level drives a higher expression of Fas and, thus, greater susceptibility to apoptosis [4].

HSPA1A/Hsp70 (heat-shock protein A1A/70)

Serum concentrations of serum heat-shock protein 70 (Hsp70/HSPA1A) are strongly correlated with the markers of hemolysis (total bilirubin level, LDH activity, plasma free hemoglobin level) and hepatocellular damage (ASAT and ALAT activity) [45,54]. Researchers have noted a relationship between platelet count and serum Hsp70 levels [4,45]. Elevated Hsp70 concentration reflects inflammation, oxidative stress, and liver injury. Extracellular Hsp70 derived from damaged necrotic cells promotes a proinflammatory (Th1) immune response, which can play a role in developing a maternal systemic inflammatory response [45]. This results in endothelial damage noticed in HELLP syndrome. Hsp70 serum levels are significantly higher in preeclamptic and HELLP women compared to normal controls [4].

PP 13 (placental protein 13)

Placental protein 13 (PP 13) or galectin-13 is exclusively synthesized by syncytiotrophoblast cells [46] and is detectable in maternal blood as early as 5 weeks gestation age [62]. The syncytiotrophoblast membrane found at the interface between maternal and fetal blood flow has an abnormal morphology of its brush border in HELLP syndrome [46,63]. Recent data have revealed impaired incorporation in this membrane of placental protein 13 [6,63].

Recent research on galectin-13 has revealed an altered concentration of this protein in pregnant women affected by preeclampsia or gestational diabetes. This molecule is considered a potent immunomodulator, and it regulates T-cell function in the placenta. A recent study cited that PP13 diminishes the apoptosis rate in neutrophils and elevates HGF (hepatocyte growth factor), TNFα, ROS (reactive oxygen species), and MMP-9 (matrix metallopeptidase 9) production in these cells. A significant role of PP13 represents the regulation of neutrophil activity in the placenta by polarizing them toward a placental-growth-permissive phenotype [64].

LSECs are fenestrated endothelial cells. They allow passive molecule exchange between the hepatocytes and the sinusoids of the liver [65]. They play a role in the regulation of sinusoidal blood flow and liver regeneration, and they are involved in hepatic complications [65]. Endothelial transport takes place through fenestrae of the LSECs, but it can also occur through endocytosis and transcytosis [4]. Normal fenestrae prevent free passage to the space of Disse of soluble molecules, lipoproteins, virus particles, chylomicrons, or other nanoparticles with a diameter more prominent than the fenestrae [4,65].

Paracrine and autocrine cell signaling is required to maintain LSEC and LSEC fenestration [65]. VEGF plays an important role in this process [65]. VEGF stimulates NO release from NO synthase (eNOS) and prostacyclin release, as well as increases activin A [65]. Recent studies have shown an increased VEGF level but a decreased VEGF signaling in HELLP patients [6,20,21]. sFlt1, a soluble form of VEGFR, is elevated in sera of HELLP women. This reduces activation of VEGFR in LSEC. Furthermore, an increase in sEng in HELLP syndrome was demonstrated. It caused less activation of TGFRII, followed by a diminished release of VEGF and prostacyclin and less vasodilation [6,21,26]. Galectin-1 release from the placenta is increased in the serum of HELLP women to compensate for this antiangiogenic status. Galectin-1 is a soluble proangiogenic factor that enhances activation of VEGFR by binding to neuropilin-1 (NRP-1) and causes activation of platelets. Remarkably, disaccharide lactose completely blocks the activation of platelets [27,28,29]. Thus, there is a reduction in the overall activation of VEGFR in LSECs. All the above result in decreased fenestration of LSECs by activating hepatic stellate cells following sinusoidal capillarization, reduced recruitment of bone marrow-derived progenitor cells, and increased apoptosis of LSECs. Bone marrow-derived progenitor cells play a role in LSEC regeneration [65,66]. Sinusoidal capillarization involves less filtration of chylomicron remnants [65]. It leads to postprandial hypertriglyceridemia, an elevated risk of atherosclerosis, and an increased fibrotic state of the liver [65]. Restoring VEGF homeostasis resolves intra- and extrahepatic abnormalities seen in HELLP syndrome [66].

The fenestrae of LSECs show a dynamic change in diameter. The ability to adapt is altered by sinusoidal blood pressure, hormones, drugs, toxins, extracellular matrix changes, aging, and exposure to environmental pollutants [50]. Elevated blood pressure determines dilation of fenestrae [50]. Gal-1 causes calcium mobilization from the extracellular space to the cytosol [4,50]. The fenestral diameter is regulated by calcium through the activation of the calcium–calmodulin–actomyosin pathway. An increased intracellular calcium level causes contraction of fenestrae [65,67]. A reduction in intracellular calcium concentration in patients treated with prostaglandin E1 has been noted, leading to the dilation of fenestrae [67]. In HELLP syndrome, decreased VEGFR signaling promotes an elevated ET-1 serum level and, thus, a reduction in the diameter and number of fenestrae [4,67].

LSECs act as a sphincter by swelling or contracting in response to vasoactive substances. The narrowing of the sinusoidal lumen limits blood flow [30,31]. Leukocytes might pile up at these narrowing sites, causing blood flow obstruction. Usually, red blood cells flow easily through such narrow points unless the lumen is reduced to near zero [4]. In HELLP patients, this narrowing is close to maximum because of an abundance of vasoactive substances [4]. As a result, ischemia and coagulation are promoted [4].

Toxins can cause LSEC swelling and gap formation by damaging the fenestrae [37]. Thus, the sinusoids disintegrate, and blood flow is reduced. This mechanism is implicated in sinusoidal obstruction syndrome (SOS) [4]. Galactosamine acts as a toxin, causing gap formation and LSEC swelling [37]. An elevated activin B serum concentration has been noted in HELLP women [37]. Activin B by itself proves no harm to the liver considering its weak expression and failure to inhibit DNA synthesis in hepatocytes [4]. A higher activin B expression results in higher FSH release. FSH contains galactosamine. Therefore, an elevated FSH level might be responsible for the increased exposure of LSECs to galactosamine [40,65]. This could result in LSEC necrosis [40]. NO, MMP-2, and MMP-9 inhibitor limit LSEC injury [65]. Inhibition of these substances promotes injury [65]. Inhibition of NO and MMP-9 inhibitors is seen in HELLP syndrome. Gal-1 stimulates the expression of MMP-9 and CXCL16. An elevation in sFlt1, sEng, and ADMA concentration leads to NO decrease [26,33]. As a result, increased injury in LSECs and an inflammatory state promoted by CXCL6 have been observed [4,27].

Activin A and VEGF act in a synergic manner. Both contribute to the LSEC survival by maintaining the extracellular matrix [4]. Activin A stimulates collagen production in stellate cells and tubulogenesis in LSECs [38]. Prior studies remarked a decreased activin A and VEGF level in HELLP women but a higher inhibin A level [4,42]; it was also observed that inhibin A behaves like an activin A inhibitor [37]. These molecules lead to an increased apoptosis rate of LSECs and a diminished restoration [4].

In the sera of HELLP patients, the FasL level is elevated [4,60]. FasL binds Fas-positive cells to induce apoptosis through an intrinsic pathway or CD4^+^ cell activation [4,60]. A higher expression of Fas is seen in LSECs and hepatocytes [4]. Although the liver produces Fas, the expression of FasL was not detected in HELLP women [4]. Therefore, the placenta seems to be the source of Fas-induced apoptosis of hepatocytes and LSECs in the liver [4]. Cardier et al. [61] reported that Fas ligation usually does not lead to apoptosis on its own in LSECs. These cells are more resistant to the Fas-induced apoptosis pathway. TNFα stimulates the susceptibility of LSECs by activating these cells and increasing Fas expression [61]. TNFα does not promote apoptosis by itself. In HELLP syndrome, the TNFα level is elevated [6]. In HELLP women, TNFα induces activation of LSECs followed by increased apoptosis by activating the Fas receptor. The FasL form seen in HELLP patients is a multimeric, highly active form with a molecular weight between 66 and 150 kDa [42]. This particular form of FasL cannot pass the fenestrae of LSECs. Apoptosis of hepatocytes takes place when LSECs degrade [4,42].

Hsp70 is a cytokine that indicates tissue damage and is expressed by stressed and necrotic cells [45]. This substance can evoke a proinflammatory (Th1) immune response. It can activate and increase the apoptosis rate of endothelial cells [45]. Angs-2 can also activate endothelial cells and exert an inflammatory state [51]. Angs-2 inhibits Angs-1 activity, which promotes cell survival, sprouting, tube formation, and quiescence of endothelial cells [51]. Within HELLP sera, elevated levels of both Hsp70 and Angs-2 have been noticed, leading to more robust endothelial activation. As a result, endothelial cells are more susceptible to undergoing apoptosis [39,53,63].

## 3. Genetic Studies

To this day, research has not been able to identify a worldwide genetic cause for an increased risk for HELLP syndrome. A probable etiological mechanism is represented by the combined effect of multiple gene variants (Table 2) with additional effects of maternal and environmental factors. Recent studies have stated that gene variants in Fas and VEGF gene and coagulation factor V Leiden (FVL) mutation are associated with an increased risk of HELLP compared to healthy women [10,65]. Variants in the glucocorticoid receptor gene and Toll-like receptor gene increase the risk of HELLP syndrome significantly more than PE [6].

## 4. Pathogenetic Mechanisms in HELLP Women

### 4.1. The Inflammatory Response

Compared to a normal pregnancy, the inflammatory response is more enhanced in HELLP patients. It is vital to always keep in mind that fulminant disseminated intravascular coagulation (DIC) may develop super acutely in HELLP syndrome. The inflammatory response with coagulation cascade and complement activation is caused by syncytiotrophoblast particles (STBM) and other placental factors [66,67]. These substances interact with maternal immune cells and vascular endothelial cells [6].

CRP, interleukin 6, and TNFα serum levels are elevated in HELLP women [37]. White blood cell counts are higher in these patients and are correlated with syndrome severity [72]. Impaired complement regulation might lead to the development of thrombotic microangiopathy seen in HELLP patients [73].

Active multimeric von Willebrand factor (VWF) serum level is higher in HELLP syndrome compared to normal pregnancy [74]. It is released by strongly activated vascular endothelial cells, and it promotes platelet aggregation and favors platelets adherence to vessel intima [6].

### 4.2. Thrombotic Microangiopathy

Damage to vascular endothelial cells produced by antiangiogenic factors, exposure to TNFα, and high levels of active VWF seen in HELLP syndrome may interact and result in thrombotic microangiopathy [68,70]. The active multimeric form of VWF is depolymerized in circulation by metalloproteinase ADAMT13 [6]. A higher concentration of active VWF is seen in HELLP due to a decreased serum level of ADAMT13 in these patients. A condition termed catastrophic antiphospholipid syndrome (APLS) develops in women with APLS and HELLP [6]. This affliction ends in extensive microangiopathy and multiorgan failure [6].

### 4.3. Microangiopathic Hemolytic Anemia

Erythrocytes are damaged as they pass through blood vessels with impaired endothelium and fibrin strands, resulting in microangiopathic hemolytic anemia (MAHA) [6]. A transient indicator could be an abnormal peripheral blood smear with schizocytes and/or burr cells [7]. Lactate dehydrogenase (LDH) concentration may rise as a consequence of hemolysis. Free hemoglobin binds to unconjugated bilirubin in the spleen or haptoglobin in plasma [7]. Low serum haptoglobin is a characteristic in HELLP women [16]. The substances resulting from intravascular hemolysis promote the activation of coagulation cascade and increase DIC risk [6].

### 4.4. Liver and Kidney Dysfunctions

Placenta-derived FasL (CD95L) is toxic to human hepatocytes [42]. A higher concentration of FasL is seen in the villous trophoblast and maternal blood in HELLP [75]. FasL triggers TNFα production, thus inducing apoptosis and necrosis of the hepatocytes [76]. Intense staining with TNFα and elastase antibodies is seen in this condition. Autopsies revealed hepatocyte necrosis with no fatty cell transformation, surrounded by fibrin strands and hemorrhages, with rare subcapsular bleeding and infarcts [77]. As a result of thrombotic microangiopathy, fibrin and leukostasis have been observed in sinusoids [78]. The microangiopathy seen in HELLP syndrome enhances hepatocyte damage because it significantly restricts portal blood flow [79].

Kidney dysfunction is usually moderate in HELLP patients. A possible cause might be glomerular endotheliosis [6]. A renal biopsy performed in women with HELLP syndrome and postpartum renal failure revealed thrombotic microangiopathy and acute tubular necrosis [6].

### 4.5. Disseminated Intravascular Coagulation (DIC)

The main activator of coagulation is the tissue factor (TF) [80]. Fetal microparticles mimic TF activity. TF could be exposed to injured vascular endothelial cells [6]. Its activation is enhanced by activated platelets and increased levels of coagulation factors, and it is promoted by thrombotic microangiopathy [6]. Coagulation inhibitors inactivate these factors [6]. In patients with severe HELLP and multiorgan failure, the serum concentrations of thrombin–inhibitor complexes are high, suggesting an exacerbated activation of coagulation. Fibrin and platelet aggregates appear in circulation, platelets and inhibitors are consumed, and DIC is present. A coexistence of HELLP and placental abruption has been cited [6]. In this particular case, blood clots and thrombin enter maternal circulation, causing systemic defibrination [6].

Compensate DIC resembles moderate consumption of platelets and inhibitors, with overt bleeding rarely appearing. Therefore, this condition hardly affects the prognosis. In women where increased activation is only partly compensated, thrombotic microangiopathy may be associated [6]. This clinical condition can rapidly progress to manifest DIC with bleeding from the skin and mucous membranes and often multiorgan failure [6].

Laboratory findings to diagnose DIC are represented by platelet count, D-dimer, antithrombin, protein C, thrombin–antithrombin complex (TAT), and prothrombin time. The criteria to diagnose overt or uncompensated DIC requires several abnormal findings in these parameters [6].

D-dimer serum level is elevated in nearly all women with a HELLP pregnancy in compensated DIC [6]. A moderately elevated D-dimer concentration is a nonspecific indicator of disease activity. As a consequence of angiopathy and liver dysfunction, decreased values of platelet count, antithrombin, and protein C might be seen in HELLP [6]. Compensated DIC may be well present with high D-dimer levels [6].

Only one of the conventional laboratory tests may specifically reflect the DIC process, i.e., the TAT assay [16]. A value over 10 μg/L is suggestive for DIC [16]. Unfortunately, valuable data from the TAT assay in HELLP patients are not cited enough in studies. It seems that adding the TAT assay to platelet count, D-dimer, and antithrombin is a valuable asset for better monitoring these women [6].

## 5. Conclusions

Our review aimed to discuss a few hypotheses of the complex pathophysiology of HELLP syndrome. The placenta-derived factors might have a significant influence on LSECs. Considering their function, we presume they are responsible for LSEC capillarization with fibrosis, resulting in diminished LSEC renewal, contraction of fenestrae, and swelling of LSECs. This obstructs sinusoids and limits blood flow, promoting activation and apoptosis of LSECs. The cascade of events eventually impedes sinusoidal blood flow, causing hepatocyte decay, which ultimately leads to liver failure.

## Figures and Tables

**Figure 1 medicina-58-00326-f001:**
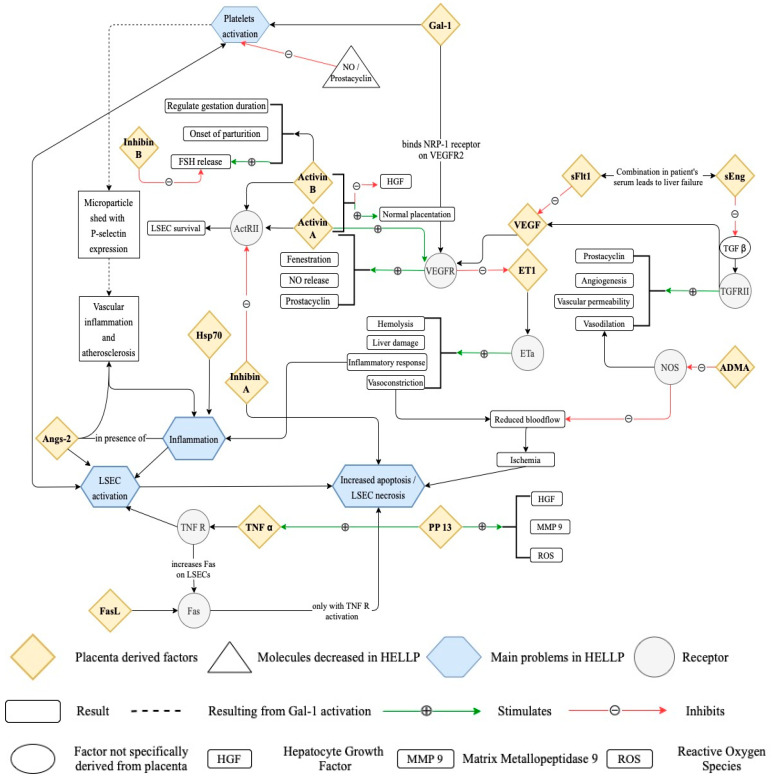
Placenta-derived factors involved in HELLP pathology.

**Table 1 medicina-58-00326-t001:** Placental factors involved in pathogenesis of liver in HELLP women.

Factors	HELLP	Role	References
Factors influencing the vasculature			
VEGF	↑	Proangiogenic, prevents hypertension	Abildgaard et al. [6], Bussen et al. [20], Purwosunu et al. [21]
sFlt1	↑↑	Antiangiogenic, ⊖ vasodilation, ⊕ ET-1	Abildgaard et al. [6], Zhou et al. [22], Tache et al. [23], Whitehead et al. [24], Schaarschmidt et al. [25], Venkatesha et al. [26], Purwosunu et al. [21]
sEng	↑↑	⊖ TGFβ signaling, ⊖ vasodilation, antiangiogenic, influences vascular permeability	Abildgaard et al. [6], Venkatesha et al. [26], Purwosunu et al. [21]
Gal-1	↑	Proangiogenic, matrix remodeling, procoagulant	Freitag et al. [27], Schnabel et al. [28], Pacienza et al. [29]
ET-1	↑↑	Promotes hypertension, increases hemolysis, liver enzymes, CD4^+^ and CD8^+^, decreases platelets	Morris et al. [30], Karakus et al. [31]
Angs-2	↑↑	Activates endothelial cells	Karakus et al. [31], Torry et al. [32]
ADMA	↑	⊖ NO synthase	Siroen et al. [33], Savvidou et al. [34]
Growth factors			
Inhibin A	↑	Antagonise activin A	Mylonas et al. [35], Muttukrishna et al. [36]
Activin A	0	⊖ Mitogen-induced DNA synthesis, ⊕ apoptosis in hepatocytes, ↑ tubulogenesis of LSECs induced by VEGF, ↑ survival of LSECs	Mylonas et al. [35], Endo et al. [37], Rodgarkia-Dara et al. [38], Florio et al. [39], Muttukrishna et al. [36]
Inhibin B	0	↓ FSH release	Mylonas et al. [35], Vassalli et al. [40], Cook et al. [41]
Activin B	↑	↑ FSH release, promotes labor	Mylonas et al. [35], Vassalli et al. [40], Muttukrishna et al. [36]
Apoptosis/necrosis-related factors			
FasL/Fas	↑↑	⊕ Apoptosis	Abildgaard et al. [6], Strand et al. [42], Gibbens et al. [43]
Hsp70	↑↑	Marker of tissue damage, ⊕ proinflammatory immune response, ⊕ endothelial injury	Molvarec et al. [44,45]
PP 13	↑	Development of fetal/maternal interface, immune regulation	Than et al. [46]

VEGF—vascular endothelial growth factor; sFlt1—soluble fms-like tyrosine kinase 1; sEng—soluble endoglin; Gal-1—galectin-1; ET-1—endothelin-1; Angs-2—angiotensin-2; ADMA—asymmetric dimethylarginine; ↑—higher than pregnant controls; ↑↑—higher than ↑; 0—no significant difference compared with controls; ⊕—stimulates; ⊖—inhibits; ↓—decreases.

**Table 2 medicina-58-00326-t002:** Genetic variants associated with an elevated risk of developing HELLP syndrome.

Gene Variant	HELLP Compared to	Outcome	References
Glucocorticoid receptor gene (GCCR)	Healthy pregnant	Abnormal immune and glucocorticoid	Abildgaard et al. [6]
Bell SNP polymorphisms	Severe PE	sensitivity	
Toll-like receptor 4 gene (TLR4)	Healthy pregnant	Inflammation	Van Rijn et al. [68]
D299G	PE	Ineffective immunity	
T3991			
Polymorphisms			
VEGF gene (VEGFA)	Healthy pregnant	Angiogenesis, vasculogenesis	Nagy et al. [69]
C−460T	Healthy pregnant	arterial muscular relaxation	
G+405C			
Polymorphisms			
Fas (TNFRSF6) gene, homozygous	Healthy pregnant	Immune regulation, apoptosis	Sziller et al. [70]
Polymorphism in A−670G		Liver disease	
FV Leiden	Healthy pregnant	Thrombophilia	Muetze et al. [71]

## Data Availability

All information included in this review is documented by relevant references.

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
