# Peer review of "HELLP Syndrome—Holistic Insight into Pathophysiology"

_medicina, 2022, doi:10.3390/medicina58020326_

Round 1

Reviewer 1 Report

The manuscript is well structured. The content has an adequate order and its presentation is fortunate. Bibliographic citations are correct.

Author Response

The manuscript is well structured. The content has an adequate order and its presentation is 
fortunate. Bibliographic citations are correct.

Response: Thank you for the appreciative words.

Reviewer 2 Report

This review by Petca et al explores the pathophysiology of HELLP syndrome. I have some suggestions for improvement:

It may be worth rewording the first sentence of the abstract so that HELLP is clearly defined.  

Throughout, some words here are not used correctly in this context. For example, in the abstract it says “…a previous HELPP pregnancy is at high risk of developing this entity…” In addition to correcting these, the whole manuscript would benefit from editing to improve grammar and sentence structure.

Some references, for example ref 2 and associated statistics (0-24% mortality rate) are not clearly represented. Perhaps this just needs to be reworded. Also there are more recent works that show morbidity and mortality rates of HELLP that may be better suited.

Particularly since this is a review, more detail should be provided on certain points. In the introduction, inflammatory involvement and inheritability is mentioned but not discussed at all.  

In discussing the factors influencing the vasculature, they are all briefly mentioned and introduced. Again since this is a review, rather than a dot point format, it may be worth exploring these factor in more detail and include more current research in these areas.

Points are not particularly well explored. The content is, but it would be useful for the authors to further explain what a concept suggests and how it may contribute to the broader topic.  

“We” is used often. Although I understand its use in some circumstances, I would be cautious as it can sound like the authors are claiming the work at times.

The conclusion nicely states the authors view on HELLP and their hypothesis of its etiology and pathophysiology. This does not come across throughout the manuscript. It also may be worth rewording the aim of the review in the abstract.

Author Response

This review by Petca et al explores the pathophysiology of HELLP syndrome. I have some 
suggestions for improvement:
It may be worth rewording the first sentence of the abstract so that HELLP is clearly defined. 
Response: Thank you for the suggestion. We made the necessary change – see line 22. 
Throughout, some words here are not used correctly in this context. For example, in the abstract 
it says “…a previous HELPP pregnancy is at high risk of developing this entity…” In addition to 
correcting these, the whole manuscript would benefit from editing to improve grammar and 
sentence structure.
Response: We made the necessary adjustment, thus, improving the clarity - see line 25.
Some references, for example ref 2 and associated statistics (0-24% mortality rate) are not clearly 
represented. Perhaps this just needs to be reworded. Also there are more recent works that show 
morbidity and mortality rates of HELLP that may be better suited.
Response: Thank you for the thorough read of our article. It is a pertinent observation, and we verified 
the statistics and added two more references for that particular statement.
Particularly since this is a review, more detail should be provided on certain points. In the 
introduction, inflammatory involvement and inheritability is mentioned but not discussed at all. 
Response: Thank you for the constructive remarks and significant input that improved our paper.
Reviewer 2 also suggested and we have gathered data on the inflammation mechanisms and illustrated 
them in Figure 1. Please see lines 99-101.
In discussing the factors influencing the vasculature, they are all briefly mentioned and 
introduced. Again since this is a review, rather than a dot point format, it may be worth exploring 
these factor in more detail and include more current research in these areas. Points are not 
particularly well explored. The content is, but it would be useful for the authors to further explain 
what a concept suggests and how it may contribute to the broader topic. 
Response: Thank you for this remark. We inserted more data to clarify activins and PP 13 roles further 
– Please see lines 202 -205, 237, 242-249. We also added three more references – 59, 62, and 64.
“We” is used often. Although I understand its use in some circumstances, I would be cautious as 
it can sound like the authors are claiming the work at times.
Response: Thank you for your attention. As suggested, the “we”s were removed from the article.
The conclusion nicely states the authors view on HELLP and their hypothesis of its etiology and 
pathophysiology. This does not come across throughout the manuscript. It also may be worth 
rewording the aim of the review in the abstract.
Response: Thank you for the significant input that improved our paper. The Conclusions section 
tone was adapted accordingly with a review article voice.

Reviewer 3 Report

This paper is an extended review about pathophysiology of HELLP syndrome. The authors give a complete overview of the different pathogenic mechanisms involved in this serious complication (inflammatory response, thrombotic microangiopathy, microangiopathic hemolytic anemia, liver and kidney dysfunctions, disseminated intravascular coagulation). Moreover, the authors clarify the pathogenic pathways that end up causing liver dysfunction (placental factors and vasoactive substances induce an inflammatory state and endothelial damage, resulting in sinusoidal obstruction syndrome (SOS) that leads to an increased formation of microthrombi, ischemia of the hepatocytes, and finally liver failure).

In order to a better understanding of the patogenic pathways, I just suggest you add a figure/scheme reflecting how the placental factors and vasoactive substances induce the inflammatory state and endothelial damage and finally liver failure.

Author Response

This paper is an extended review about pathophysiology of HELLP syndrome. The authors give
a complete overview of the different pathogenic mechanisms involved in this serious complication 
(inflammatory response, thrombotic microangiopathy, microangiopathic hemolytic anemia, liver 
and kidney dysfunctions, disseminated intravascular coagulation). Moreover, the authors clarify 
the pathogenic pathways that end up causing liver dysfunction (placental factors and vasoactive 
substances induce an inflammatory state and endothelial damage, resulting in sinusoidal 
obstruction syndrome (SOS) that leads to an increased formation of microthrombi, ischemia of 
the hepatocytes, and finally liver failure).
Response: Thank you for the thorough read of our article. 
In order to a better understanding of the patogenic pathways, I just suggest you add a 
figure/scheme reflecting how the placental factors and vasoactive substances induce the 
inflammatory state and endothelial damage and finally liver failure.
Response: Thank you for an excellent suggestion that, we consider, consistently improved our article. 
We have gathered data on the inflammation mechanisms and illustrated them in Figure 1. Please see 
lines 99-101.

Round 2

Reviewer 2 Report

The authors have made adequate changes to the manuscript.